# Astrocytes in Neurodegenerative Diseases: A Perspective from Tauopathy and α-Synucleinopathy

**DOI:** 10.3390/life11090938

**Published:** 2021-09-09

**Authors:** Peng Wang, Yihong Ye

**Affiliations:** Laboratory of Molecular Biology, National Institute of Diabetes and Digestive and Kidney Diseases, National Institutes of Health, Bethesda, MD 20892, USA; wang.peng@nih.gov

**Keywords:** neurodegenerative disease, Alzheimer’s disease, Parkinson’s disease, astrocyte, tauopathy, α-synucleinopathy, Tau, α-synuclein, cell-to-cell transmission, prion

## Abstract

Neurodegenerative diseases are aging-associated chronic pathological conditions affecting primarily neurons in humans. Inclusion bodies containing misfolded proteins have emerged as a common pathologic feature for these diseases. In many cases, misfolded proteins produced by a neuron can be transmitted to another neuron or a non-neuronal cell, leading to the propagation of disease-associated pathology. While undergoing intercellular transmission, misfolded proteins released from donor cells can often change the physiological state of recipient cells. Accumulating evidence suggests that astrocytes are highly sensitive to neuron-originated proteotoxic insults, which convert them into an active inflammatory state. Conversely, activated astrocytes can release a plethora of factors to impact neuronal functions. This review summarizes our current understanding of the complex molecular interplays between astrocyte and neuron, emphasizing on Tau and α-synuclein (α-syn), the disease-driving proteins for Alzheimer’s and Parkinson’s diseases, respectively.

## 1. Introduction

Neurodegeneration refers to the progressive loss of structure and function of neurons in pathological conditions. Depending on the type and location of the affected neurons, neurodegenerative diseases can display heterogeneous clinical and pathological expressions [1]. Although research in the past has long been ‘neurocentric’, recent studies have started to shift the paradigm as new roles by glial cells in neurodegenerative diseases are being revealed. 

Glial cells were first reported in 1856 by a pathologist named Rudolf Virchow in the book ’Cellular Pathology’. Derived from the ancient Greek word “glía” (meaning “glue” in English), the name “Glia” suggests these cells as “glue” that holds neurons together. However, this view has changed significantly in recent years as more and more neuronal supporting functions were identified for glial cells.

Glial cells are historically categorized into two main groups: macroglia and microglia. The former includes astrocytes, oligodendrocytes, NG2-glia, and ependymal cells, while microglia are resident phagocytes of the central nervous system (CNS). Among these cell types, astrocytes have drawn significant attention recently due to their unique neuron-safeguarding functions. As the most abundant non-neuronal cells in the CNS, astrocytes are capable of responding to many neurodegeneration-associated events such as metabolic fluctuation, molecular damage, and energy and ion homeostasis disruption [2]. Additionally, as immune-responding cells, astrocytes also participate in neuroinflammation [3]. These functions are all tightly regulated during ageing and ageing-associated neurodegeneration. Here, we review the emerging roles of astrocytes in two major pathological conditions, tauopathies and α-synucleinopathies.

## 2. Astrocytes in Tauopathies

### 2.1. Tau and Tauopathies

Intracellular neurofibrillary tangles (NFTs) formed by hyperphosphorylated Tau are a pathological hallmark of a broad spectrum of neurodegenerative disorders collectively referred to as tauopathies [4,5,6]. Tauopathies are conventionally classified into two groups. Primary tauopathies, which include progressive supranuclear palsy (PSP), frontotemporal dementia parkinsonism linked to chromosome17 (FTDP-17), Pick’s disease (PiD), corticobasal degeneration (CBD), chronic traumatic encephalopathy (CTE), and argyrophilic grain disease (AGD), refer to disease conditions in which Tau deposit is the predominant pathology [4,7]. By contrast, secondary tauopathies involve other pathogenic drivers in addition to Tau deposition. For example, Alzheimer’s disease (AD), the most prevalent cause of dementia, is a secondary tauopathy because it also involves extracellular deposition of amyloid-β (Aβ) plaques [8,9]. 

Tau is a microtubule-binding protein predominantly expressed in neurons in the brain [10,11]. However, Tau deposits are prevalent in both neuronal and non-neuronal cells in tauopathies. Immunohistochemistry analyses of phosphorylated Tau revealed six distinct astroglial phenotypes associated with tauopathies including astrocytic plaques (AP), tufted astrocytes (TA), ramified astrocytes (RA), and globular astroglial inclusions (GAI) in primary tauopathies, and thorn shaped astrocytes (TSA) and granular/fuzzy astrocytes (GFA) in aging-related Tau astrogliopathy (ARTAG) [12,13,14,15].

The expression of Tau is regulated by alternative splicing of the Tau-encoding gene MAPT [16]. The resulting six isoforms contain either 3 or 4 microtubule-binding repeats (referred to as 3R and 4R, respectively) combined with zero to two amino-terminal insertions (NT). Healthy adults express approximately equal amounts of 3R- and 4R-Tau, and aggregates composed of either 3R or 4R Tau have been seen in different tauopathies. However, sporadic tauopathies such as PSP, CBD, FTDP-17, and AGD feature NFT deposits exclusively composed of 4R-Tau [14]. 

Post-translational modifications (PTMs) of Tau such as phosphorylation, acetylation, ubiquitination, SUMOylation, methylation, and glycation have long been recognized as a critical contributing factor to tauopathies [17,18,19,20,21]. Tau PTMs may enable the formation of the highly ordered β-sheet structures, which facilitates the formation of filamentous Tau inclusions, as indicated by a recent study that reported a role of Tau ubiquitination in filament formation and strain specification [22]. PTMs may also control Tau stability, and thus influence Tau pathology, as exemplified by the implication of ubiquitin ligase and deubiquitinase (DUB) in Tau stability regulation [23,24,25]. Among reported PTMs, Tau hyperphosphorylation is thought to be the most significant driving force of tauopathy, possibly because this modification changes the affinity of Tau to microtubule, and thus its aggregation propensity. Noticeably, Tau phosphorylation was also seen in astrocytes, implying a potential role in reactive astrogliosis [26].

### 2.2. Astrocytes as a Modulator of AD and Tauopathies

Although most tauopathies including late-onset AD-associated tauopathies arise sporadically within the population, genome-wide association study (GWAS) have identified many tauopathy-associated single-nucleotide polymorphism (SNP) markers [27,28,29]. Intriguingly, many genes associated with increased risk of neurodegeneration are glial genes (Table 1).

ApoE is the strongest genetic risk locus for AD. ApoE E4 carriers have enhanced AD pathology, accelerated cognitive decline and worsened memory performance compared to noncarriers [30]. As a secreted lipid transport protein that moves lipids between organs, ApoE is expressed primarily in a subset of astrocytes in the CNS [31,32]. The mechanism by which ApoE variants alter AD pathology is complex, which is likely linked to the deposition and clearance of Aβ in the brain [33,34,35,36,37].

Given the tight link between AD and tauopathy, the role of ApoE in tauopathies has also been examined. By crossing the P301S Tau transgenic mice to those bearing a human ApoE knock-in allele or lacking ApoE completely, Shi et al. showed that P301S/ApoE E4 mice had significantly higher levels of intracellular Tau, more microglia activation and reactive astrogliosis compared to P301S mice bearing other ApoE variants, while the P301S mice lacking ApoE completely had the least tauopathy [38]. More recently, the same group found that astrocyte specific removal of ApoE E4 allele markedly decreased phosphorylated Tau and Tau-associated neurodegeneration, which suggested that astrocyte-derived ApoE4 is a major regulator of tauopathies [61]. However, another study suggested that neuronal ApoE expression is linked to MHC-I upregulation, which causes tauopathy and selective neurodegeneration [62].

CLU gene variants (encoding ApoJ/Clusterin) are another strong genetic risk factor for late-onset AD, as established by GWAS [40,41]. Like ApoE, CLU is an apolipoprotein predominantly expressed in astrocytes in the brain [63]. As an extracellular chaperone, CLU secreted by astrocytes can bind to Aβ to prevent Aβ aggregation [64,65,66]. Accordingly, it has been proposed that increased CLU in glia may be protective in AD and tauopathies [67].

Other AD risk factors identified by GWAS include FERMT2 (encoding Kindlin-2) [43] and WWOX [29]. FERMT2 is mainly expressed in astrocytes [68] but can also be detected in human induced pluripotent stem cell (iPSC)-derived neurons [45]. It is localized to focal adhesions where it interacts with and activates β3 integrin [69]. The role of FERMT2 in AD and tauopathy is largely unknown, but a genome-wide siRNA screen suggested that FERMT2 may increase Aβ peptide production by elevating the levels of mature APP at the cell surface via membrane recycling [44]. Another candidate-based screening found that knockdown of FERMT2 led to a reduction of phosphorylated Tau [45]. WWOX, encoding a putative oxidoreductase, is expressed in both astrocytes and neurons [29]. WWOX regulates Aβ aggregation and also binds to Tau to influence Tau hyperphosphorylation and neurofibrillary formation [47,48]. Taken together, these genome-wide studies not only identified genetic risk factors for AD and related tauopathies, but also underscored a role for glial cells, especially astrocytes, in driving Aβ- and Tau-associated neuropathology.

### 2.3. Astrocyte in the Propagation of Tauopathies 

An unusual characteristic of tauopathies is the prion-like propagation of Tau-containing fibrils, which correlates with cognition decline and disease progression. Braak and colleagues first reported the spatial and temporal dynamics of Tau-containing fibrils in AD brains. Specifically, NFTs, first uncovered in the transentorhinal region, appear to traverse along several anatomical paths to reach the hippocampus and eventually the neocortex region [70]. The progressive spreading of Tau inclusions was later recapitulated in mouse models [71,72,73,74]. There is now comprehensive evidence that supports the idea that pathogenic Tau species undergo cell-to-cell transmission with a prion-like property [75,76,77,78]. However, the ultimate spatial distribution of Tau NFTs is distinct among tauopathies due to strain distinctions. Additionally, external factors may also influence the spreading pattern of tauopathy. For example, in AD, genetic and clinical evidence indicates that Aβ plaque deposition can facilitate the spreading of tauopathy [79,80,81]. Moreover, Tau-containing aggregates accumulated in glial cells (both microglia and astrocytes) may also modulate Tau transmission (see below).

The intercellular transmission of Tau is likely initiated when neurons release Tau either in monomeric or small oligomerized forms. Indeed, Tau is readily detected in the interstitial fluid (ISF) of the brain under normal conditions [82]. Accumulating evidence suggests that Tau species can be released from neurons independent of cell death, and this process is modulated by neuronal activities [83,84,85]. The mechanisms underlying Tau release are controversial. Specifically, some studies showed that Tau is predominantly released in a free soluble form [86,87,88,89] but other studies suggested membrane-associated vesicles such as exosome as an extracellular Tau carrier [90,91]. It is possible that multiple mechanisms coexist to regulate Tau secretion. 

Once in the cell exterior, Tau may be taken up by cells via endocytosis [92], micropinocytosis [93] or other forms of cargo internalization [94]. One study suggests that healthy neurons efficiently take up both normal and aggregated Tau by distinct but overlapping mechanisms, which indicates the existence of multiple Tau receptors for internalization [95]. Not only neurons, but other cell types in the brain such as microglia and astrocytes can also engulf Tau proteins [39,93,96]. In certain immortalized cells, endocytosis of Tau preformed fibrils (PFFs) is initiated when Tau binds to the cell surface heparan sulfate proteoglycans (HSPGs) [94,97,98], which cooperate with a membrane receptor to mediate Tau internalization [99]. However, HSPG does not play a major role in Tau uptake in primary astrocytes [99,100]. We recently used a spatially resolved proteomic mapping strategy to identify the integrin αV/β1 complex as a receptor that binds human Tau fibrils to mediate their entry into astrocytes [39]. When inside the astrocyte, Tau may be cleared by lysosomal degradation or the recently reported astrocytic glymphatic system [101]. 

Although Tau aggregates have been observed in various cell types in the brain, most attention in the field has been given to intraneuronal or extracellular Tau deposits, while the glial involvement was rarely considered. This deficiency may significantly hinder our understanding of the mechanisms underlying the transmission of tauopathy. To better understand the role of glial Tau deposits in tauopathy, the following questions need to be carefully addressed. (i) Which glial cell type accumulates the most pathological Tau in tauopathies? (ii) Which Tau species is propagated in each tauopathy and how is their distribution in the brain sculpted? (iii) Do astrocytes or other glial cells contribute to Tau propagation? (iv) Does the accumulation of Tau in astrocytes contribute to neurodegeneration, and if so, what is the underlying mechanism?

To date, only a few published studies attempted to address these questions, which collectively paint an incomplete model. Tau accumulation in astrocytes was reported in some tauopathy mouse models [102,103]. More recently, using an in vivo reporter system, Anastasie et al. demonstrated bidirectional exchanges of Tau protein between neuron and astrocyte. They further showed that soluble Tau, but not Tau aggregates, is toxic to a subpopulation of hippocampal astrocytes [104]. This study hints at a role for astrocytes in tauopathy. A few studies investigated the disease relevance of astrocytic Tau in other experimental models. For example, expression of human Tau in glia in a Drosophila model led to neurotoxicity, suggesting that Tau, if propagated into glial cells, might have a pathogenic activity [105]. Likewise, in a transgenic mouse model, astrocyte-specific expression of human Tau leads to neurodegeneration [106]. A study by Richetin et al. also suggests astrocytic Tau as a causal factor for dementia. They detected Tau accumulation in astrocytes of the hilus, a portion of the hippocampus in AD patients; in mice, overexpression of the 3R Tau variant in hilar astrocytes of the dentate gyrus impaired mitochondrial function and thus ATP production [107]. Intriguingly, this work detected 3R Tau in astrocytes, unlike previous studies that attributed astrocytic Tau deposits predominantly to the 4R isoform [108]. 

Two recent papers further link Tau to the build-up of astrocytic senescent cells in the brain, which contribute to neurodegeneration. Musi et al. showed that destroying senescent cells in mice at early stages of tauopathy slows neurodegeneration and corrects aberrant brain blood flow [109], whereas Bussian et al. reported that specific elimination of senescent astrocytes is sufficient to prevent neurodegeneration and cognitive decline in a mouse model of tauopathy [110]. Although these studies both hinted at a critical role for astrocytic Tau in cell senescence, which in turn influences neurodegeneration, how the senescent state of microglia or astrocytes is aligned with other tauopathy-related features remains unclear. Altogether, the existing evidence suggests that in tauopathies, Tau proteopathy may exist beyond neurons, which warrants additional studies.

### 2.4. Tauopathies Are Associated with Widespread Reactive Astrogliosis

Under neurodegeneration conditions, astrocytes also undergo significant changes, which can fall into three morphologically defined categories: (i) atrophy/degeneration occurs as astrocytes lose their homeostatic function to support neuronal growth. (ii) Astroglial remodeling refers to morphologic alterations of astrocytes under disease or CNS injury conditions. (iii) Reactive astrogliosis refers to special responses of astrocytes to different insults in many CNS disorders, which result in astroglial hypertrophy (increased volume, thickened processes, and increased expression of GFAP etc. [111,112]). 

Due to their sensitivity to the brain environment, astrocytes can enter a “reactive” or “activated” state now generally termed astrogliosis [113]. Many markers of reactive astrocytes [2,114,115] have been identified and used to characterize the neurodegenerative disease states. Under certain experimental conditions, reactive astrogliosis induced by lipopolysaccharide (LPS) increases the phagocytic activity of astrocytes, which may mitigate tauopathies if the activated astrocytes help to clear protein aggregates [116,117]. However, reactive astrogliosis under pathophysiological conditions can also be a major contributor of chronic neuroinflammation (Figure 1), which exacerbates neurodegeneration in several animal disease models [118,119]. Thus, it seems that upon activation, astrocytes might be transformed into multiple functional states, resulting in a heterogeneous population.

A recent transcription profiling study identified two gene expression signatures corresponding to two functional states of reactive astrogliosis termed as A1 and A2, respectively. A2 astrocytes have a neuron-supporting function and can restore neuronal activities after injury. By contrast, A1 astrocytes not only fail to promote synapse formation, but also release some neurotoxic factors. Complement C3 was later identified as an astrocyte-released factor that induces neuronal impairment, possibly through a C3 receptor (C3aR) because C3aR1 deficiency reverses plaque-proximal synapse loss in a Tau P301S mouse model [120]. Interestingly, astrocyte-released C3 appears to crosstalk to microglia as well, indicating a possible vicious cycle among neuron, astrocyte, and microglia in tauopathy [121].

The A2 to A1 switch of astrocytes, instigated by microglia, appears to convert astrocytes from a supporter of neuronal homeostasis to a cell death promoter in AD. Interleukin-1α (IL-1α), tumor necrosis factor α (TNFα), and complement component 1q (C1q) secreted from activated microglia were shown to collectively induce the A1 switch phenotype [122]. Hence, blocking the transformation of astrocytes to the A1 state by these factors may be a potential therapeutic strategy, as suggested by a recent study using a Parkinson’s disease (PD) mouse model [123]. However, this oversimplified model has recently been challenged. Concerns were raised regarding the potential overlook of astrocytic heterogeneity and the complexity of the factors implicated in shaping the astrocyte phenotypes during disease progression [124,125]. Recent advancements in single cell transcriptomics may help better define the various astrocytic states associated with different pathological conditions [126,127]. 

One of the common insults that change astrocyte state in neurodegenerative diseases is abnormal protein aggregates such as Aβ-, Tau-, and α-syn-containing fibrils. For instance, Aβ peptides, derived from abnormal processing of amyloid precursor protein (APP), can form distinct aggregated states, which activate different astrocytic receptors to induce a pro-inflammatory NFκB pathway [128,129]. Distinct Tau species also differentially activate integrin signaling in primary mouse astrocytes, which leads to NFκB activation and the release of pro-inflammatory cytokines and chemokines [39]. In rodent models of AD and Huntington’s disease (HD), NFκB activation in astrocytes was observed [119,130]. Thus, the NFκB pathway appears to be a critical link that connects extracellular proteotoxic insults to astrogliosis and neuroinflammation.

Besides the NFκB pathway, the Janus kinase/signal transducer and activator of transcription 3 (JAK/STAT3) pathway is also ubiquitously involved in cell proliferation, survival, and differentiation. STAT3 was recently suggested as a mediator of reactive astrogliosis under pathological conditions such as AD and HDs [131]. However, the contributions of STAT3-mediated reactive astrogliosis to these diseases are not entirely clear. For example, one study suggested that JAK/STAT3 activation is associated with a scar-forming astrocyte activity in a model of acute spinal cord injury [132], which limits inflammation spreading [133]. By contrast, in an APP/PS1 model of AD, STAT3 deficient animals showed reduced β-amyloid levels and plaque burden, decreased pro-inflammatory cytokines, and rescued memory decline. Similarly, in a Tau mouse AD model, inhibition of STAT3 also rescues Tau pathology, ameliorates neuroinflammation, and reverses synaptic deficits [120]. Thus, whether reactive astrogliosis is detrimental or beneficial for damaged neurons may depend on the cause of neurodegeneration.

## 3. Astrocytes and α-Synucleinopathies

### 3.1. α-Synuclein and Parkinson’s Disease

α-synuclein (α-syn) was originally identified as a protein recognized by an antiserum against purified cholinergic synaptic vesicles [134]. It was localized to a patch of nuclear membrane extended to the presynaptic nerve terminal, which gave rise to the name of synuclein (protein present in synapse and nucleus). In the CNS, α-syn is predominantly associated with a pool of synaptic vesicles [135,136] where it appears to regulate a SNARE complex [137,138] and hence the membrane fusion events in endocytosis and exocytosis [139]. Accordingly, it may regulate synaptic plasticity and neurotransmitter release [140,141,142]. Additionally, α-syn contains a lipid binding domain essential for membrane association, lipid packing, and membrane remodeling [143,144]. Lastly, a chaperone activity was demonstrated in vitro using high concentrations of recombinant α-syn, but the physiological relevance of this finding remains unclear [145].

Although the precise function of α-syn is unclear, numerous lines of evidence have linked its dysfunction to PD pathology. These include (1) the existence of familial PD cases caused by duplication or triplication of the α-syn gene [146], (2) the identification of mutations in α-syn gene that are causally linked to PD [147,148], (3) the α-syn-containing intraneuronal Lewy bodies (LB) or Lewy neurites (LN) are a major pathologic hallmark of PD and a heterogeneous group of disorders referred to as α-synucleinopathies [149], and (4) a close correlation of α syn oligomerization and fibril propagation with the PD progression [150,151].

The mechanism of Lewy body formation remains elusive. The cellular α-syn concentration is likely a key determinant because in animals, the fibrillization kinetics of α-syn are highly sensitive to the protein concentration [152,153,154,155]. For example, genetic alterations in other PD-associated genes such as leucine-rich repeat kinase 2 (LRRK2) and glucocerebrosidase (GBA) can influence lysosome-mediated α-syn turnover to promote its aggregation [156,157]. Likewise, GWAS has identified additional variations in genes associated with lysosomal degradation pathways such as CHMP2B, TMEM175, SCARB3 and BAG3, which also influence α-syn aggregation [158]. In addition to genetic factors, many aging-associated events such as mitochondrial dysfunction [159], oxidative and ER stress [160,161] can all impact endogenous α-syn levels to affect the build-up of α-syn aggregates. Besides α-syn concentration, PTMs likely also regulate its aggregation. These include phosphorylation, acetylation, and protease-mediated cleavage [162,163]. Most studies along this line were conducted either using recombinant proteins in vitro or in cell lines. However, a recent report showed that α-syn cleaved by matrix metalloproteinases is modified by a glutaminyl cyclase at the N-terminus. The resulting pyroglutamate79-α-syn, which was detected in patient Lewy bodies, is more prone to form toxic oligomers [164].

### 3.2. Intercellular Transmission of α-Syn in α-Synucleinopathies

Several lines of evidence suggested that α-syn can also propagate from neuron to neuron in a prion-like manner, which can be recapitulated by cell-based models and in animals inoculated with preformed α-syn fibrils. The accumulation of α-syn-containing inclusions has also been seen in non-neuronal cells across the spectrum of α-synucleinopathies. Clinical studies showed that the topographical distribution of astrocytic α-syn inclusions closely mirrors that of the cortical intraneuronal LN and LB in PD [165], suggesting that the build-up of astrocytic α-syn aggregates may contribute to the symptomatic progression of these diseases. However, subcortical astrocytes in multiple-system atrophy (MSA) and corticobasal degeneration (CBD) do not accumulate α-syn aggregates [166]. The heterogeneity of astrocytic α-synucleinopathies in various PD variants suggest the involvement of multiple factors besides distinct conformational strains of α-syn in α-synucleinopathies, which await further elucidation [167,168,169].

Since α-syn is predominantly expressed in neurons, the observed α-syn-containing inclusions in astrocytes suggest that astrocytes scavenge extracellular α-syn released from neurons via active endocytosis [170,171,172], which should reduce extracellular α-syn and thus, the inter-neuronal transmission of α-Syn (Figure 1). Additionally, α-syn is efficiently transferred from astrocyte to astrocyte, but less efficiently from astrocyte to neuron [173]. Thus, it is unlikely that astrocytes can serve as a mediator in α-syn propagation between neurons. Altogether, these findings suggest that astrocyte might promote the elimination of α-syn inclusions in early phases of α-synucleinopathies, which offers a beneficial effect. However, persistent accumulation of α-syn aggregates might overload astrocytes, inducing stress phenotypes to impair astrocyte functions.

### 3.3. Role of Astrocyte in α-Syn-Associated Neuroinflammation

Like tauopathies, a key aspect of PD pathology is neuroinflammation, which is associated with reactive astrogliosis and the loss of dopaminergic neurons in the substantia nigra pars compacta (SNc) [174,175,176,177]. Inflammation-associated cytokines like interleukin-1 β (IL1-β), interleukin-6 (IL6), and tumor necrosis factor-α (TNFα) are all elevated in the cerebrospinal fluid (CSF) or serum of PD patients [178,179,180]. This neuroinflammation response used to be considered as an event downstream of dopaminergic neuron loss. However, emerging evidence suggests that disturbed astrocytes may play an active role in α-syn-associated neuroinflammation prior to the loss of dopaminergic neurons. 

It turns out that extracellular α-syn aggregates can directly interact with astrocytes via a pattern recognition receptor such as Toll-like receptor (TLR) 4, inducing a TLR4-dependent inflammatory response [181,182]. Moreover, the induction of pro-inflammatory cytokines and chemokines correlates well with the level of astrocytic α-syn, suggesting that the neuron-to-astrocyte transmission of α-syn aggregates may be coupled to neuroinflammation [171]. α-syn also strongly upregulates IL6 and inflammatory mediator intercellular adhesion molecule-1 (ICAM-1) in human astrocytes and in a human U-373 MG astrocytoma cell line, which is further linked to the activation of the major mitogen-activated protein kinase (MAPK) pathway [183]. Cytokines released by activated astrocytes can induce neuronal death, but the underlying mechanisms remain unclear [184]. When PD-related A53T mutant α-syn was expressed in astrocytes in a mouse model, increased accumulation of α-syn aggregates in astrocytes was found in pre-symptomatic and asymptomatic mouse brains, correlating with the expansion of reactive astrogliosis. These mice also developed progressing paralysis before the onset of PD-like symptoms. This study argued for a critical involvement of astrocytic α-syn in neurodegeneration via a cell non-autonomous mechanism [185].

## 4. Conclusions and Perspectives

In summary, neurodegenerative diseases are not a disease of one cell type. Although neuronal cell death is the primary cause of disease symptoms, the underlying mechanisms are complex and can be influenced by both neuronal factors as well as non-autonomous factors from other cell types. Astrocytes, being the most abundant non-neuronal cells in human brains, can play a significant role in neurodegenerative diseases. More attention should be given to research along this direction. Many outstanding questions need to be addressed by both in vitro cell-based assays and animal models. For example, what controls the switch that changes astrocytes from a neuron supporter to a death promoter? Are misfolded Tau or α-syn by itself sufficient to activate astrocytes in vivo? What is the precise astrocyte-activating conformation of misfolded proteins? What are the membrane receptors that recognize these misfolded proteins? Are there functionally distinct populations of activated astrocytes, and if so, what is the population that causes neuronal cell death? What are the toxic factors released from astrocytes? Is there a way to maintain the astrocytes’ garage-cleaning function without stimulating them into an inflammatory state? Answers to these questions will fill an important gap in our understanding of these devastating diseases.

## Figures and Tables

**Figure 1 life-11-00938-f001:**
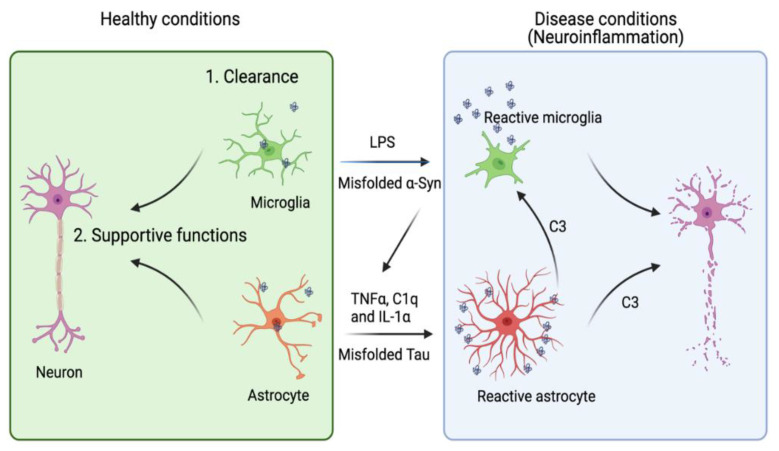
A dual role of microglia and astrocytes in neuronal growth and neurodegenerative diseases. Accumulating evidence suggests that the activation of microglia and astrocytes may be a double-edged sword. Under healthy conditions, microglia and astrocytes engulf neuron-derived misfolded proteins such as Tau and α-syn to promote protein homeostasis in the brain micro-environment. Astrocytes can also provide other supportive functions including axonal guidance and synaptic support. However, when these cells are overactivated by toxic factors (e.g., LPS or excess amount of extracellular Tau or α-syn), they release pro-inflammatory cytokines and chemokines to disrupt neuronal integrity. Reactive microglia can also cross-activate astrocytes by releasing cytokines such as TNFα, IL1-α and C1q. Conversely, astrocytes release complement C3, which can act on both microglia and neurons to further enhance neuroinflammation. Image created in BioRender.com.

**Table 1 life-11-00938-t001:** A list of astrocyte- or microglia-specific AD and tauopathy modulators.

Gene	Glia Cell Type	Pathway	Effect on Aβ	Effect on Tau
APOE [30]	Astrocyte	Lipid metabolism, immune response	Aβ clearance [34]	Tau aggregation and toxicity [38,39]
CLU(APOJ) [40,41]	Astrocyte	Lipid metabolism, immune response	Amyloid formation [42]	Unknown
FERMT2 [43]	Astrocyte	Integrin signaling, and cell adhesion, angiogenesis	Aβ production [44]	Tau proteostasis [45]
WWOX [29]	Astrocyte	Putative oxidoreductase, neuronal differentiation [46]	Aβ aggregation [47]	Tau phosphorylation, NFT formation [47,48]
IL1RAP [49]	Astrocyte, oligodendrocyte	Neuronal synaptogenesis [50]	Unknown	Unknown
PTK2B [51]	Microglia, astrocyte	Immune response, endocytosis, synaptic transmission	Unknown	Tau toxicity [52]
SORL1 [53]	Microglia, astrocyte	Endosomal traffic	APP trafficking [54]	Unknown
CELF1 [55]	Astrocyte, oligodendrocyte, microglia	Unknown	Unknown	Unknown
EPHA1 [56,57]	Astrocyte, oligodendrocyte, microglia	Cell migration and proliferation, immune response	Unknown	Tau toxicity [52]
CD2AP [56,57]	Astrocyte, oligodendrocyte, microglia	Neurite structure modulation and blood-brain barrier integrity	Aβ production [58,59]	Tau toxicity [60]

## Data Availability

Not applicable.

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
