# Peer review of "Astrocytes in Neurodegenerative Diseases: A Perspective from Tauopathy and α-Synucleinopathy"

_life, 2021, doi:10.3390/life11090938_

Round 1

Reviewer 1 Report

This is a comprehensive and timely review on the role of astrocytes in neurodegenerative diseases with focus on tauopathy and alpha-synucleinopathy. It is well-structured and concisely written and of high importance for readers in the field.

Minor points that should be addressed:

(i) Figure 1: Please use consistent style: 1. Clearance 2. Supportive functions. Also, in the "Disease conditions" panel, "pro-inflammatory" or "neuroinflammation" should be added.

(ii) Lines 293-294: A recent publication on a novel a-synuclein post-translational pyroglutamate modification should be cited (PMID: 34309760).

Author Response

We would like to thank the reviewer for his/her support.  We have changed figure 1 according to the suggestion.  We also add a sentence summarizing the recent discovery on pyroglutamate-modification of cleaved synuclein. The changed text is highlighted in blue.  

Reviewer 2 Report

The review by Wang and Ye is summarizing some recent knowledge on the contribution of astrocytes to tauopathy and synucleinopathy. The text is well written and interesting but lacks some recent information.

Regarding AD, the authors cite sometimes known data on Aß. It would be clearer to either strictly focus on tauopathy or treat both Tau and Aß.

Figure 1 shows that reactive astrocytes secrete C3 but the bibliography on this data is not cited and the figure is unclear. According to the literature, the C3 pathway between astrocytes and microglia influences neuronal tauopathy.

There are also data concerning the role of the astrocytic glymphatic system mediated by aquaporin 4 and the accumulation of Tau.

It would be interesting to precise which cells express MAPT (line 66) in the brain and at which stage (if it is known) and to cite more precisely the post-translational modifications of Tau contributing to neurodegeneration (line 73).

There is now a consensus statement regarding the way to cite astrocyte reactivity which is “reactive astrogliosis” (Escartin et al. 2021).

Some important astrocytic pathways involved in reactive gliosis and that contribute to AD (ex STAT3 …) are not discussed.

Line 186: replace “In AD patients” by “in AD patients”

Author Response

We thank the reviewer for the positive comments. 

1. Regarding AD, the authors cite sometimes known data on Aß. It would be clearer to either strictly focus on tauopathy or treat both Tau and Aß.

Answer: We shortened the section by deleting some sentences regarding Abeta.  However, given the tight connection between Abeta and Tau deposition, we feel that it is impossible to completely avoid mention Abeta.

2. Figure 1 shows that reactive astrocytes secrete C3 but the bibliography on this data is not cited and the figure is unclear. According to the literature, the C3 pathway between astrocytes and microglia influences neuronal tauopathy.

Answer: We have changed the figure accordingly.  The missing citation is not included. Ref. 106, 107.

There are also data concerning the role of the astrocytic glymphatic system mediated by aquaporin 4 and the accumulation of Tau.

Answer: We thank the reviewer for this excellent suggestion. However, due to space limitations, we cannot elaborate too much on this interesting new finding.  We decide to include a recent review on this topic as reference 87.

It would be interesting to precise which cells express MAPT (line 66) in the brain and at which stage (if it is known) and to cite more precisely the post-translational modifications of Tau contributing to neurodegeneration (line 73).

Answer: We have added a few references to address this point. Ref. 11 and 16. We also expand the discussion on Tau PTMs.

There is now a consensus statement regarding the way to cite astrocyte reactivity which is “reactive astrogliosis” (Escartin et al. 2021).

Answer: We have now changed the term to "reactive astrogliosis" throughout the paper to comply with this new guideline.

Some important astrocytic pathways involved in reactive gliosis and that contribute to AD (ex STAT3 …) are not discussed.

Answer: We added a new paragraph in section 1.4 to discuss the role of STAT3 in Tauopathy.

Line 186: replace “In AD patients” by “in AD patients”

Answer: changed.